# Identification of Rubber Plantations in Southwestern China Based on Multi-Source Remote Sensing Data and Phenology Windows

Guokun Chen [1,2], Zicheng Liu [1,*], Qingke Wen [3], Rui Tan [1], Yiwen Wang [1], Jingjing Zhao [1] and Junxin Feng [1]

1   Faculty of Land Resource Engineering, Kunming University of Science and Technology, Kunming 650093, China
2   Key Laboratory of Plateau Remote Sensing, Yunnan Provincial Department of Education, Kunming 650093, China
3   National Engineering Research Center for Geomatics (NCG), Aerospace Information Research Institute, Chinese Academy of Sciences, Beijing 100101, China
*   Correspondence: liuzc@stu.kust.edu.cn

**Abstract:** The continuous transformation from biodiverse natural forests and mixed-use farms into monoculture rubber plantations may lead to a series of hazards, such as natural forest habitats fragmentation, biodiversity loss, as well as drought and water shortage. Therefore, understanding the spatial distribution of rubber plantations is crucial to regional ecological security and a sustainable economy. However, the spectral characteristics of rubber tree is easily mixed with other vegetation such as natural forests, tea plantations, orchards and shrubs, which brings difficulty and uncertainty to regional scale identification. In this paper, we proposed a classification method combines multi-source phenology characteristics and random forest algorithm. On the basis of optimization of input samples and features, phenological spectrum, brightness, greenness, wetness, fractional vegetation cover, topography and other features of rubber were extracted. Five classification schemes were constructed for comparison, and the one with the highest classification accuracy was used to identify the spatial pattern of rubber plantations in 2014, 2016, 2018 and 2020 in Xishuangbanna. The results show that: (1) the identification results are in consistent with field survey and rubber plantations area generally shows a first increasing and then decreasing trend; (2) the Overall Accuracy (OA) and Kappa coefficient of the proposed method are 90.0% and 0.86, respectively, with a Producer's Accuracy (PA) and User's Accuracy (UA) of 95.2% and 88.8%, respectively; (3) cross-validation was employed to analyze the accuracy evaluation indexes of the identification results: both PA and UA of the rubber plantations stay stable over 85%, with the minimum fluctuation and best stability of UA value. The OA value and Kappa coefficient were stable in the range of 0.88–0.90 and 0.84–0.86, respectively. The method proposed provides reliable results on spatial distribution of rubber, and is potentially transferable to other mountainous areas as a robust approach for rapid monitoring of rubber plantations.

**Keywords:** Google Earth Engine (GEE); identification; phenology windows; rubber plantations; random forest algorithm; sample optimization

## 1. Introduction

Rubber, a tropical evergreen broad-leaved vegetation, originates from the Amazon basin forest of South America. As the only renewable green energy material among the four major industrial raw materials, rubber is known for its economic values and carbon sequestration potential [1,2]. With the increasing demand for rubber in national defense and latex production, the planting area of artificial rubber plantations has shown a trend of substantial growth and continuous expansion in tropical rainforest areas worldwide [3–6]. As of June 2022, the Association of Natural Rubber Producing Countries (ANRPC) reported

that the global natural rubber production is 1.113 million tons, with an increase of 3.8% over the same period in 2021, while global consumption is expected to grow at a faster pace of 5.8% over the same period, to 1.206 million tons. Driven by market demand and the economy, rubber planting areas have been transplanted from traditional growing areas to and is now cultivated in almost all tropical zones (Malaysia, Laos, Myanmar, Thailand, Vietnam, Cambodia, etc.) in the past 20 years, with a transplanting area over 1 million square hectares [7–9].

Xishuangbanna Dai Autonomous Prefecture (XSBN) lies in the core area of the "One Belt and One Road" policy, and its strategic location has constantly drawn attention. Moreover, short-term economic incentives, weak enforcement of regulations and suitable climate of rainy and hot conditions make XSBN the most important rubber cultivation area in China. Meanwhile, the Sloping Land Conversion Program (SLCP) in China clearly banned slope shifting cultivation and encouraged planting of trees, which also accelerated the replacement of shifting cultivation with rubber plantations from the valleys onto progressively higher and steeper slopes, and even into the Nature Reserves. During the last decades, the rubber plantation area ratio in XSBN has climbed from 1.3% in 1976 to 22.14% in 2014, and has become the most dominant land use type in the region [10]. Compared with natural forests, commercial monoculture rubber plantations on steep highlands have many characteristics conducive to land degradation and environmental problems. First, it is more difficult for newly reclaimed rubber plantations on slopes to achieve long-term vegetation cover stability; since rubber has only one species and a simple vegetation structure, the interception of rainfall by covers becomes weak, leading to high soil erosion risk and reduced soil productivity. Second, the low water keeping capacity of the soil makes it vulnerable to climate change and human activities. Third, exotic monoculture rubber plantations are 'forests' indeed, but are intensively managed, treated with fertilizers, herbicides and fungicides and negative to biodiversity [11–13]. Despite the fact that cultivation of rubber could act as a carbon sink by sequestering carbon in biomass and indirectly in soils; however, for most cases, the sharp growth of rubber and the drastic land use change process (mainly from natural forests to rubber plantations) also resulted in a series of ecological and environmental problems, such as the weakening ecosystem deforestation, fragmentation of the remaining forest, land degradation and biodiversity loss [14,15]. Therefore, it is of great significance to acquire accurate information on the dynamic expansion process and spatial-temporal distribution pattern on rubber plantations to achieve a sustainable development goal.

Traditional monitoring approaches for rubber plantations are mainly based on field investigation, which are expensive and time-consuming, and continuous monitoring is difficult to achieve due to the poor timeliness of the data [16]. Remote sensing is a science and technology that detects, analyzes and studies the earth's resources and environment based on the interaction between electromagnetic waves and the earth's surface materials, and reveals the spatial distribution and dynamic change characteristics of various elements on the earth's surface [17]. Due to the advantages of strong timeliness and little human interference, it has been frequently employed in research such as LUCC, crop identification and vegetation phenology monitoring [18–24]. Although remote sensing technology has become a widespread approach in rubber plantations mapping [25], the majority of studies are still in the exploratory stage and the following obstacles remain unsolved: (1) Rubber is an evergreen broad-leaved vegetation, the spectral characteristics of which are easily mixed with other vegetation types such as natural forests, tea plantations, orchards and shrubs [26], and both supervised and unsupervised classification methods rely on spectral characteristics, which are fraught with uncertainty [7,27]. (2) It has been demonstrated that for rubber in the tropical northern fringe area, south of the Tropic of Cancer, a unique phenomenon of leaf fall will take place under low temperature environment during the dry season [28–30]. The emergence of this phenology feature provides a new idea for the effective identification of rubber plantations. Relevant studies have been conducted and many results have been achieved based on phenology characteristics of MODIS time



series data [31–33]. However, the coarse spatial resolution (250 m–1000 m) of MODIS data has great limitations in plateau mountainous areas with complex terrain fragmentation, and difficult to identify rubber plantations with scattered distribution [34,35]. (3) The appearance of medium and high spatial resolution remote sensing images provides a new way to identify rubber plantations. Although Landsat series images were used to establish time series data for rubber information acquisition in some studies [30,36–38], still, frequent cloudy and rainy weather greatly reduces the availability of images since most rubber plantations are distributed in tropical or subtropical rainforest areas with mixed vegetation [29,35,39]. It is unrealistic to establish long time series data only through optical images of a single sensor.

Compared with optical images, Synthetic Aperture Radar (SAR) is not affected by cloud and fog weather, possesses the properties of penetration and anti-interference, and can gather efficient ground observation data all day long [40]. Some scholars extracted forests distribution according to different polarization modes of HH (horizontal emission and horizontal reception) and HV (horizontal emission and vertical reception) between forests and other vegetation types [19,28,41–44], confirming that SAR data have significant advantages in identifying tropical forests.

At present, the emergence of the cloud computing platform represented by Google Earth Engine (GEE) breaks the traditional way of remote sensing data acquisition and preprocessing. GEE has a powerful parallel computing ability and massive online remote sensing datasets, making it possible to conduct remote sensing research of large area, long time series and high spatial-temporal resolutions [45].

Low spatial resolution remote sensing images have obvious limitations in mountainous regions with fragmented and complex terrain, making it difficult to identify rubber plantations with small planting area. Besides, affected by cloudy and rainy weather, a single satellite sensor cannot establish complete time series data. In this paper, we aim to address the above challenges of mapping rubber plantation areas in topographically and climatically complex settings. The specific objectives of this study are twofold: (1) Creating a rubber plantation map with a spatial resolution of 10 m using a pixel-based classification method integrated with phenology windows; (2) By combining the data with SAR, we hope to improve the mixing of different height vegetation (such as tea plantations and rubber plantations) and provide a reference for high-precision mapping and ecological protection of rubber plantations in southwestern China and Southeast Asian countries.

## 2. Materials and Methods

### 2.1. Study Area

Xishuangbanna Dai Autonomous Prefecture (XSBN) lies in Yunnan Province, southwestern China, 21°08′–22°36′ N, 99°56′–101°50′ E, bordering Laos in the southeast and Myanmar in the southwest, with a national border of 996.3 km long. The Lancang-Mekong River flows through XSBN from north to south, and high mountains and deep valleys characterize the whole territory [26], forming a landscape tilting from the north to the south [46]. The highest altitude is 2429 m, the lowest altitude is 477 m and the relative height difference is close to 2000 m (Figure 1). With the Lancang River as boundary, the landform structure in the eastern region is dominated by middle and low mountains and plateaus, and the remnants of the Nu Mountain Range in the west, mostly basin landforms [47].

XSBN is located on the northern border of the tropics, south of the Tropic of Cancer, and has a humid monsoon climate typical of the northern tropics. Due to the staggered influence of terrain and monsoons, the climate is divided into two distinct dry and wet seasons, but no clear four seasons. The wet season extends from late May to late October, and the dry season last from late October to late May of the following year [48–50]. The average annual rainfall ranges from 1138.6 to 2431.5 mm, with more than 80% of rainfall occurring during the rainy season [51], and the average annual temperature is between 18.9–23.5 °C.

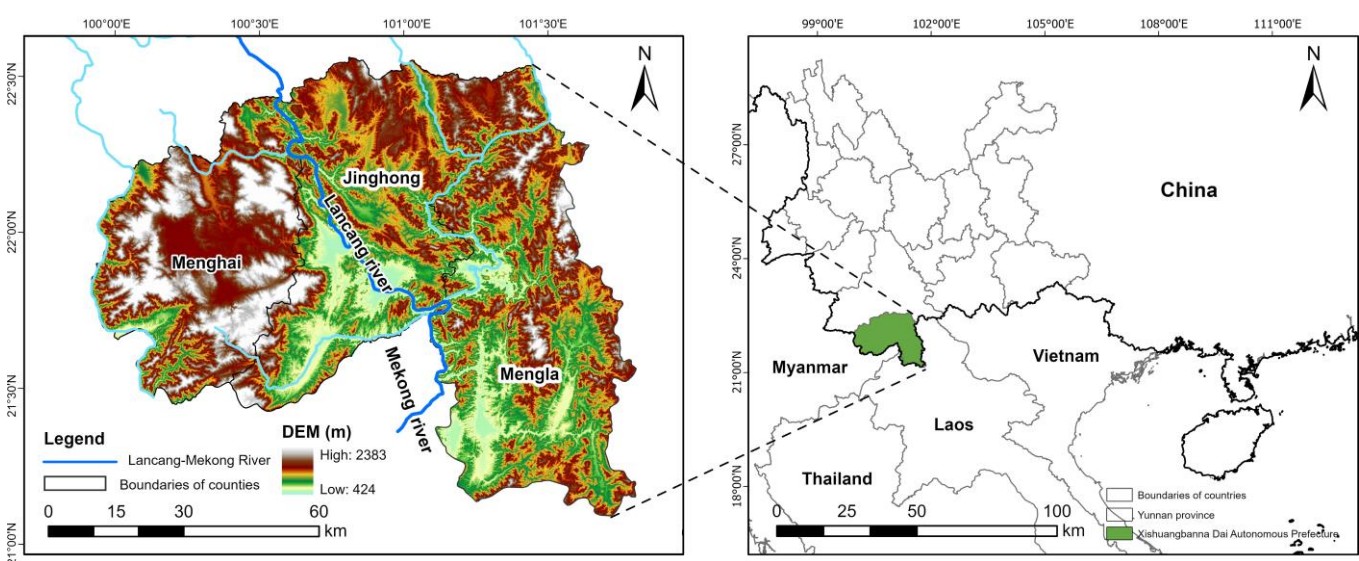

**Figure 1.** Map of XSBN, showing location, major rivers and altitude variation.

The study area is rich in biodiversity; as one of the few tropical regions in China, XSBN comprises only 0.2% of China's land area, but harbors nearly 16% of plant species, 36.2% of birds, 22% of mammals and 15% of amphibian and reptiles found in the country [52]. Distributed with the largest coverage area and the most abundant types of tropical seasonal rainforests and tropical mountainous rainforests, XSBN is also the region with the most complete preservation of tropical ecosystems in China [53]. With a total administrative area of 19,124.5 km$^2$, the state consists of one metropolis, two counties and three districts. There are 13 ethnic minorities that make up 77.9% of the state's total population, totaling 792,800 inhabitants [46].

XSBN's unique climate and geographical environment provide favorable circumstances for the development of rubber plantations. As the economic value of rubber has increased over the past decade, its cultivation area has expanded, and rubber has become the predominant land use and land cover (LULC) type and economic pillar industry in the region [26].

### 2.2. Data Sources and Preprocessing

The GEE platform was applied to acquire and pre-process the remote sensing data that were used in the study. Due to the low availability of images caused by cloudy and rainy conditions in the highland mountains, two data sources, multispectral (MSI) and synthetic aperture radar (SAR), were used to combine the advantages of various sensors in terms of temporal and spatial resolution to obtain complete time-series data. MSI data products include Landsat-7/ETM+ surface reflectance (L7_SR), Landsat-8/OLI surface reflectance (L8_SR) and Sentinel-2 MSI surface reflectance (S2_SR).

L7_SR and L8_SR are 2A-level data products obtained by LEDAPS (Landsat Ecosystem Disturbance Adaptive Processing System) and LaSRC (Landsat Surface Reflectance Code) algorithms after atmospheric correction. With a resolution of 30 m, a revisit time of 16 d, and an amplitude of 185 km, the SR data products include the whole visible (VIS), near infrared (NIR) and shortwave infrared (SWIR) spectrum. Both sets of data extend from 1999 to the present, making it possible to conduct large-scale, long-term studies for LULC classification and vegetation phenology monitoring.

S2_SR is a 2A-level SR data product obtained after atmospheric and orthographic correction, including 13 spectral bands from VIS, NIR to SWIR, and the wavelength range is from 442.3 nm to 2202.4 nm. The spatial resolution is 10 m, 20 m and 60 m, the revisit period of one satellite is 10 d, the two satellites are complementary and the revisit period is

5 d. Because of its superiority in time, spatial and spectral resolution, Sentinel-2 imagery has been broadly applied for LULC classification and other vegetation identification [23].

Benefiting from the calculation and data management mechanism of the GEE platform, the resolution matching between different data sources too much can be achieved [40]. In this study, the spatial resolution of the Landsat data is 30 m, and the GEE platform can automatically sample to 10 m to match this resolution. At the same time, the coordinate system is unified through the embedded algorithm, so that each pixel can accurately represent the same range on the ground.

In order to obtain images that can cover the entire study area, the quantity and availability of pixels in three multispectral datasets were evaluated. Specifically, (i) spatial filtering: a 10 km buffer was generated first, and all images intersecting the buffer were filtered; (ii) temporal filtering: the image availability of Landsat-7/ETM+, Landsat-8/OLI and Sentinel-2, was considered and the time period was set from 2014 to 2020; (iii) attribute filtering: based on cloudy pixels percentage and the QA60 de-clouding band, the cloud mask function was used to eliminate cloud-influenced pixels from each image, and those with acceptable quality were used in the subsequent surface parameter computation. For the three data products, the final number of accessible images (Table 1), the total number of observed pixels and the number of valid pixels (Figure 2) were determined.

**Table 1.** Total available images of Landsat-7/ETM+, Landsat-8/OLI and Sentinel-1/2 in this study.

| Dataset Year | S1_GRD | L7_SR | L8_SR | S2_SR | Total Size |
|---|---|---|---|---|---|
| 2014 | 45 | 81 | 107 | 0 | 233 |
| 2015 | 168 | 81 | 106 | 0 | 355 |
| 2016 | 198 | 86 | 109 | 0 | 393 |
| 2017 | 227 | 83 | 105 | 0 | 415 |
| 2018 | 301 | 78 | 90 | 39 | 508 |
| 2019 | 329 | 87 | 112 | 736 | 1264 |
| 2020 | 347 | 75 | 105 | 742 | 1269 |

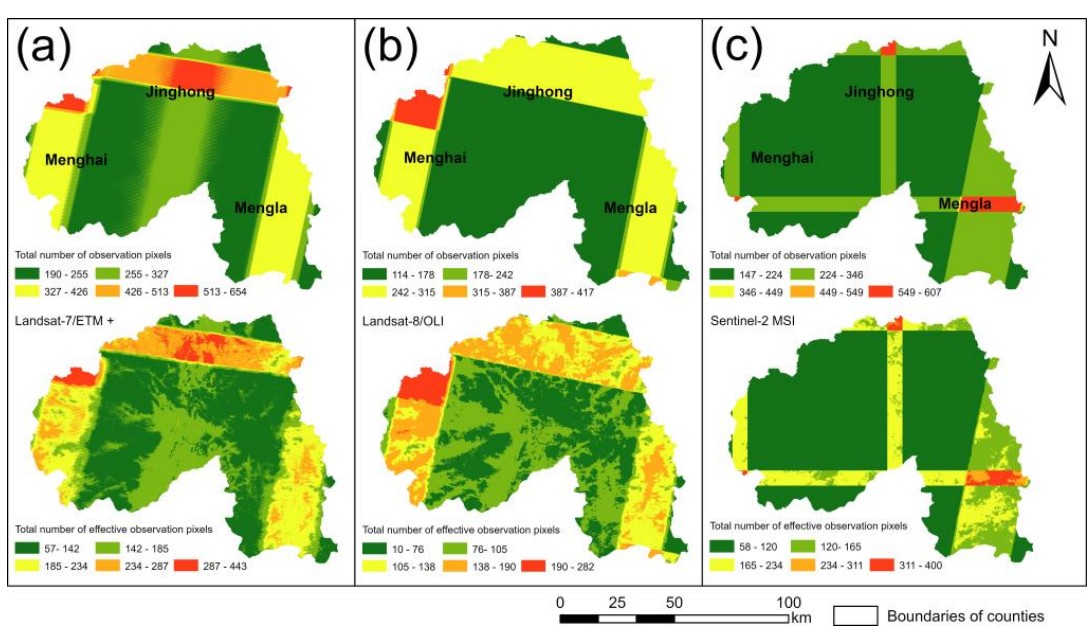

**Figure 2.** Number of total and effective pixel observations for Landsat-7/ETM+, Landsat-8/OLI and Sentinel-2 MSI in XSBN. (**a**) Landsat-7/ETM+ data from 2000 to 2020. (**b**) Landsat-8/OLI from 2013 to 2020. (**c**) Sentinel-2 MSI data from 2014 to 2020.

The SAR data employed are the Sentinel-1 SAR Ground Range Detected (S1_GRD) product, which is a first-level image dataset after Doppler Centroid Estimation, Single-Looking Composite (SLC) focusing and post-processing. There are four bands in each image, corresponding to four polarization combinations: horizontal transmit/horizontal receive (HH), horizontal transmit/vertical receive (HV), vertical transmit/vertical receive (VV) and vertical transmit/horizontal receive (VH), with a resolution of 10 m. Although SAR data were unaffected by rain and cloud cover, noise has a significant influence on data quality. We utilized the Sentinel-1 toolkit to further pre-process each scene image for speckle filtering, thermal noise removal, terrain correction and radiometric calibration [54], with ALOS 12.5 m DEM data used for the terrain correction step. The rectified images were used to generate year-by-year time series data from 2014 to 2020, with the details of each data product listed in Table 2.

**Table 2.** Detailed information of multi-sources remote sensing data used in this study.

| Sensors | Landsat-7/ETM+ | Landsat-8/OLI | Sentinel-2 A/B MSI | Sentinel-1 C-SAR |
|---|---|---|---|---|
| | L7_SR | L8_SR | S2_SR | S1_GRD |
| | Blue/450–520 nm/30 m | Blue/452–512 nm/30 m | Blue/496.6(S2A)/492.1(S2B) nm/10 m | HH/5.405 GHz/10 m |
| | Green/520–600 nm/30 m | Green/533–590 nm/30 m | Green/560(S2A)/559(S2B) nm/10 m | HV/5.405 GHz/10 m |
| **Description** | Red/630–690 nm/30 m | Red/636–673 nm/30 m | Red/664.5(S2A)/665(S2B) nm/10 m | VV/5.405 GHz/10 m |
| | NIR/770–900 nm/30 m | NIR/851–879 nm/30 m | NIR/835.1(S2A)/833(S2B) nm/10 m | VH/5.405 GHz/10 m |
| | SWIR1/1550–1750 nm/30 m | SWIR1/1566–1651 nm/30 m | SWIR1/1613.7(S2A)/1610.4(S2B) nm/20 m | — |
| | SWIR2/2080–2350 nm/30 m | SWIR2/2107–2294 nm/30 m | SWIR2/2202.4(S2A)/2185.7(S2B) nm/20 m | — |

ALOS 12.5 m DEM data (https://search.earthdata.nasa.gov/search (accessed on 1 June 2022)) are collected from the ALOS (Advanced Land Observing Satellite) satellite equipped with PALSAR sensors and a horizontal/vertical accuracy of 12.5 m that may be used for all-weather, all-day land observation. The data were uploaded to the GEE platform as one of the key features for the acquisition of information on the spatial distribution of rubber plantations in our study.

Three non-homologous LULC products were prepared including: (1) ESA_2020_10m data product jointly produced by European Space Agency produced in collaboration with a number of global research institutions, (2) ESRI_Land_Cover_2020_10m data product produced using deep learning methods by Environmental Systems Research Institute and (3) Google's near real-time 10 m resolution global LULC dataset Dynamic World generated from Tensorflow deep learning framework using GEE and AI platform, based on Sentinel-2 MSI image. The above three sets of products are used to determine the input sample type of the Random Forest classification algorithm, and the detailed information of each product is shown in Table 3.

**Table 3.** The three non-homologous LULC data products used.

| Name | Spatial Resolution | Categories | Mapping Time | Mapping Range | Mapping Accuracy |
|---|---|---|---|---|---|
| ESA_2020_10m | 10 m | 11 | 2020 | global | 74.4% |
| Esri_2020_10m | 10 m | 10 | 2020 | global | 85% |
| Dynamic World | 10 m | 9 | 2020 | global | — |

The sample data for training and validation was selected based on field investigation, Google Earth high-resolution remote sensing images, as well as the mentioned three LULC datasets. By using an integrated method of "stratified sampling + non-homogenous data voting" with visual interpretation, 6569 sample points were finally obtained, including

1051 natural forests, 843 cultivated land, 1000 tea plantations, 216 water bodies (rivers, lakes, reservoirs, etc.), 459 impervious surfaces (building lands, highways, etc.) and 3000 rubber plantations samples (Figure 3 shows a part of the sample data). All the sample points that satisfied the requirements were randomly divided into 70% training sample data and 30% validation sample data for the Random Forest classification algorithm.

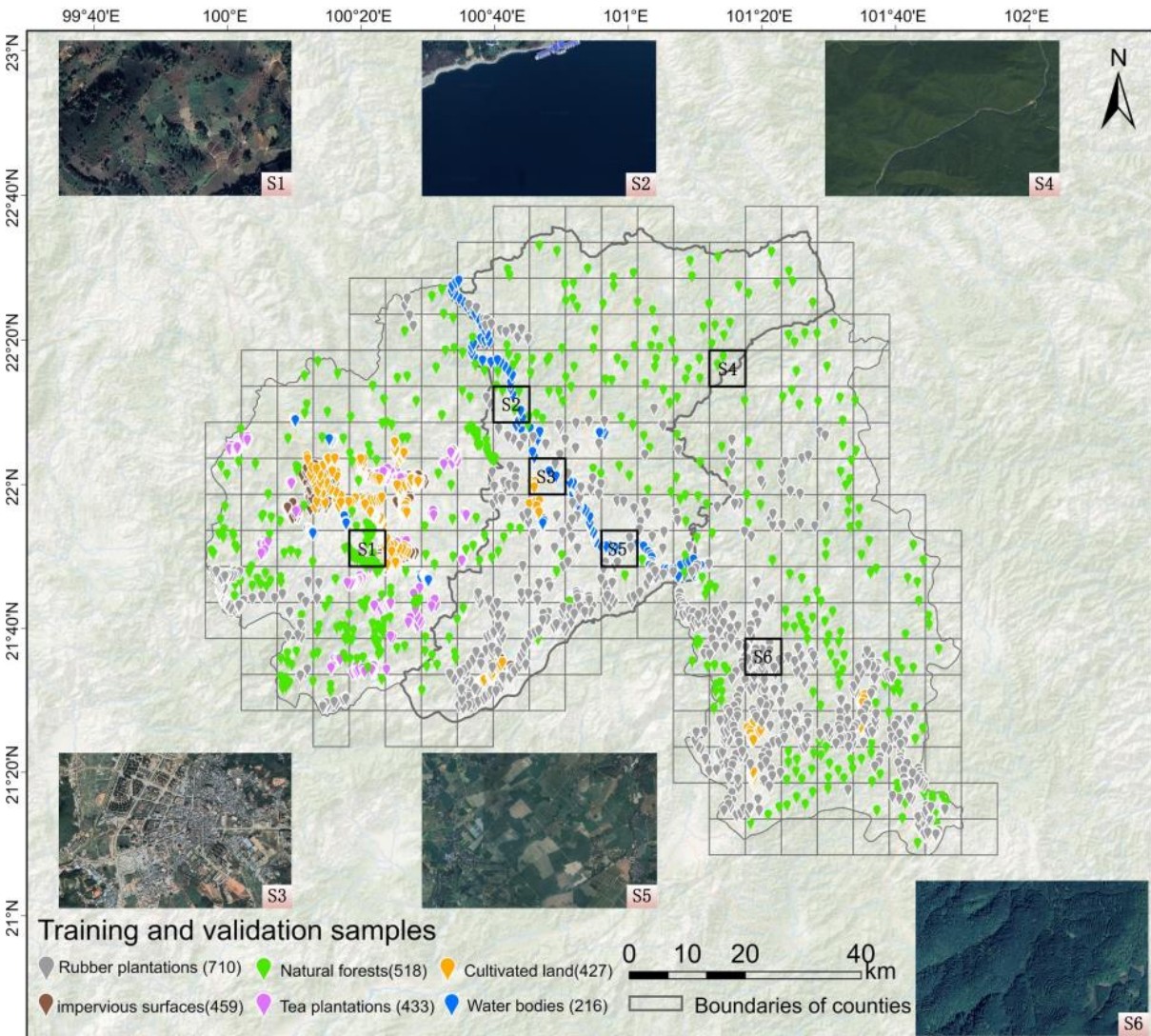

**Figure 3.** The spatial distribution of six types of training and validation samples, including (S1) tea plantations, (S2) water bodies, (S3) impervious surfaces, (S4) natural forests, (S5) cultivated land and (S6) rubber plantations, are shown in this figure. S1–S6 are the sample points collecting areas and high-resolution Google Earth remote sensing images corresponding to those areas.

The overall workflow used for rubber plantations distribution identification in XSBN is presented in Figure 4, and it consists of the following steps: (1) sample selection optimization; (2) determination of key phenology windows of rubber plantations; (3) input feature optimization; (4) classification schemes design; (5) Random Forest classification and validation; (6) accuracy assessment; (7) classification post-processing.

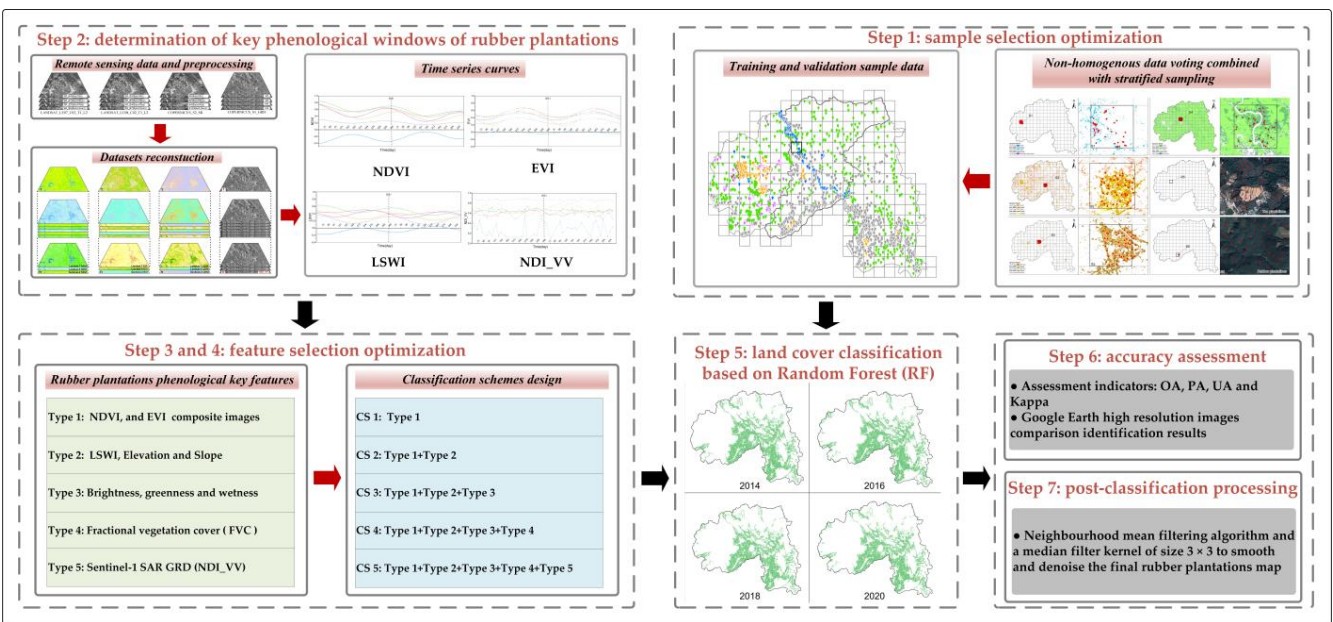

**Figure 4.** Flowchart of the remote sensing identification method proposed for rubber plantations.

### *2.3. Sample Selection Optimization*

In order to accurately identify the spatial distribution information of rubber plantations in XSBN, we optimized the method for selecting input samples. According to the actual situation of the study area, land use and land cover types were divided into six classes: natural forests, cultivated land, tea plantations, water bodies (river, lake, reservoir, etc.), impervious surfaces (construction land, road, etc.) and rubber plantations.

Non-rubber plantations sample points (with exception of tea plantations) were selected using a combination of "stratified sampling + non-homogeneous data voting" (Figure 5), with pure image pixels of natural forests, cultivated land, tea plantations, water bodies, impervious surfaces and rubber plantations classification consistent across the three data products as the selection range (that is, all three non-homologous data products considered which as the same type). We randomly generated sample points within the range.

To avoid localized clustering of sample points, the study area was gridded in blocks of 10 km × 10 km based on the GEE platform's online editing code. A fixed number of sample points were then randomly generated within each grid to ensure that the sample points were evenly distributed in the study area, and ineligible sample points within the grid were removed through visual interpretation.

As none of the above-mentioned three sets of products partitioned the tea plantations, the sample points of the tea plantations and rubber plantations were selected point-by-point in combination with Google Earth high-resolution, long time series remote sensing images as an auxiliary.

### *2.4. Determination of Key Phenological Windows of Rubber Plantations*

We chose the Normalized Difference Vegetation Index (NDVI), which can effectively reflect the density and intensity of the vegetation growth process using the calculation between NIR band and Red band [55]; the Enhanced Vegetation Index (EVI) is an optimized vegetation index that increases the sensitivity to high vegetation cover areas and enhances the ability to monitor vegetation canopy changes at the same time [55,56]; the Land Surface Water Index (LSWI) can effectively reflect plant canopy changes, soil moisture and soil surface water content status [57], whereas the NDI_VV is extremely sensitive to vegetation canopy orientation, structural changes and leaf water content [58,59]. NDI_VV, for instance, can successfully distinguish vegetation, bare ground and building land; the particular formula and explanation of each index are provided in Table 4.

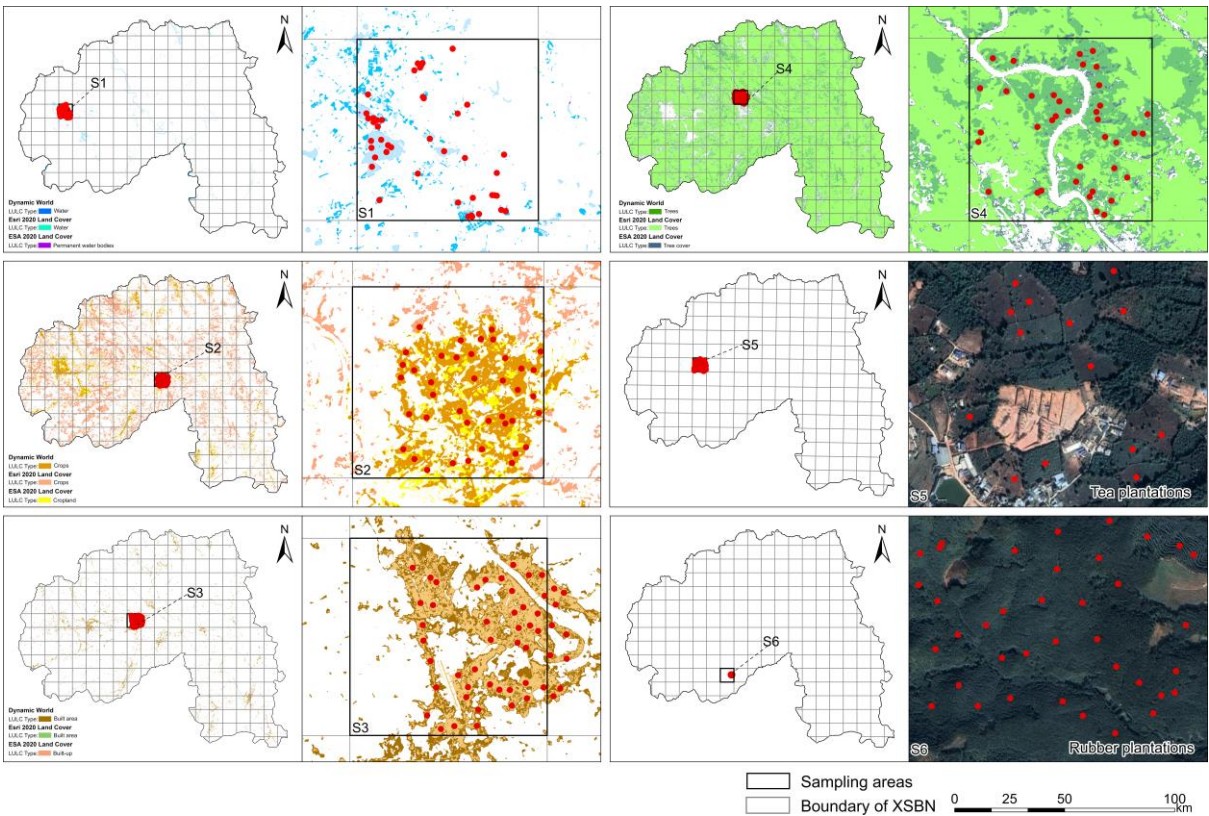

**Figure 5.** Combining stratified sampling with non-homogenous data voting, with (S1–S6) representing sampling areas for water bodies, cultivated land, impervious surfaces, natural forests, tea plantations and rubber plantations, respectively.

**Table 4.** Calculation formulas of each spectral index and phenology window period of rubber plantations.

| Indices | Expressions | Time Windows | Phenology Stages |
|---------|-------------|--------------|------------------|
| NDVI | $NDVI = \frac{\rho_{nir}-\rho_{red}}{\rho_{nir}+\rho_{red}}$ | 01/09–02/09 | Defoliation stage I |
|  |  | 02/09–02/15 | Defoliation stage II |
|  |  | 02/15–03/01 | Foliation stage I |
|  |  | 03/01–03/15 | Foliation stage II |
| EVI | $EVI = 2.5 \times \frac{\rho_{nir}-\rho_{red}}{\rho_{nir}+6.0\rho_{red}-7.5\rho_{blue}+1}$ | 05/01–05/15 | Vigorous growth stage I |
|  |  | 05/15–06/01 | Vigorous growth stage II |
|  |  | 06/01–06/15 | Vigorous growth stage III |
| LSWI | $LSWI = \frac{\rho_{nir}-\rho_{swir2}}{\rho_{nir}+\rho_{swir2}}$ | 01/09–02/09 | Defoliation stage I |
|  |  | 02/09–02/15 | Defoliation stage II |
|  |  | 02/15–03/01 | Foliation stage I |
|  |  | 03/01–03/15 | Foliation stage II |
| NDI_VV | $NDI\_VV = \frac{VV-VH}{VV+VH}$ | 02/15–03/07 | Foliation stage I |
|  |  | 03/07–03/15 | Foliation stage II |
| FVC | $FVC = \frac{NDVI-NDVI_{\min}}{NDVI_{\max}-NDVI_{\min}}$ | 01/09–02/09 | Defoliation stage I |
|  |  | 02/09–02/15 | Defoliation stage II |
| TCT | — | 01/09–02/09 | Defoliation stage I |
|  |  | 02/09–02/15 | Defoliation stage II |

On the basis of three sets of non-homologous LULC classification datasets and high-resolution Google Earth remote sensing images, 74 sampling areas were selected in the study area (Figure 6), which contains 6 of the above land use and land cover classes. Since XSBN is located in a tropical rainforest with mixed vegetation and easily affected by cloud and rain, resulting in a low number of available images in the area, it is even more difficult to ensure the quality of remote sensing images during the rainy season (the

vigorous vegetation growing season) from May to October. In order to solve the problem of incomplete multi-spectral coverage of the perennial time series data in the plateau and mountain areas, the three datasets L7_SR, L8_SR and S2_SR were reconstructed (Figure 7a) and the time series curves of NDVI, EVI and LSWI were reconstructed by a Harmonic Analysis of Time Series (HANTS) [60] (Figure 7b).

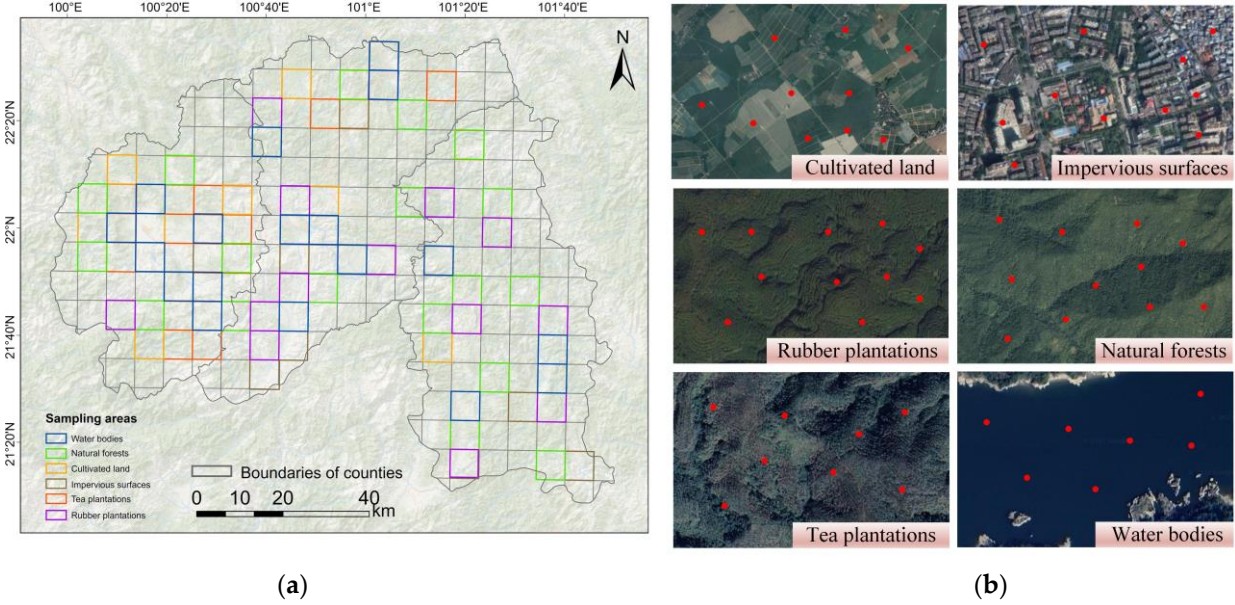

(**a**)                                                     (**b**)

**Figure 6.** Construction process of time series curves for natural forests, cultivated land, tea plantations, water bodies, impervious surfaces and rubber plantations: (**a**) selection of 74 sampling areas within the study area; (**b**) collection of sample points with high-resolution images from Google Earth.

*2.5. Feature Selection Optimization*

Figure 7b shows the annual dynamic changes of rubber plantation and other land use and land cover types in the four indexes of NDVI, EVI, LSWI and NDI_VV. Throughout the year, both natural forests and rubber plantations have high NDVI values, and their intra-annual curve variations are almost similar, showing a process of first a decline, then a rise and then a slow decline. The NDVI of natural forests dropped to the lowest value around July, and rose to the highest value around October. The NDVI values of rubber plantations, however, are second only to natural forests, and have obvious seasonal fluctuation characteristics. During the local rainy season, which lasts from late May to late October, the rubber grows vigorously and has a stable and high NDVI value; during the dry season, from November to May, the rubber enters a slow growth period and the NDVI value decreases gradually; during the dry and hot period, from early January to early February, the rubber experiences concentrated defoliation and the NDVI reaches its lowest value. From early February to mid-March, the rubber entered the new leaf germination stage, and the NDVI value began to rise. The EVI time series curves indicate that rubber has greater EVI values than natural forests throughout the year. From late January to late March, the spectral features of the two time series curves are similar; however, from late March to late December, the spectral features of the two time series curves differ significantly; this time period can be used to distinguish rubber from natural forests. The different intra-annual variability patterns of the LSWI time series curves of rubber plantations and natural forests may be utilized as an essential indication to distinguish between the two. The NDI_VV curves reveal that the natural forests curves basically do not fluctuate throughout the year, whereas the rubber and tea plantations, cultivated land and natural forests curves vary significantly from January to March. This time period can be used to differentiate between tea plantations, cultivated land and rubber plantations. The four time series curves all reflect that the spectral curves of water bodies and impervious surfaces are

significantly different from those of other land types. The spectral curves of tea plantations and cultivated land have certain distinguishability with rubber plantations in different time periods.

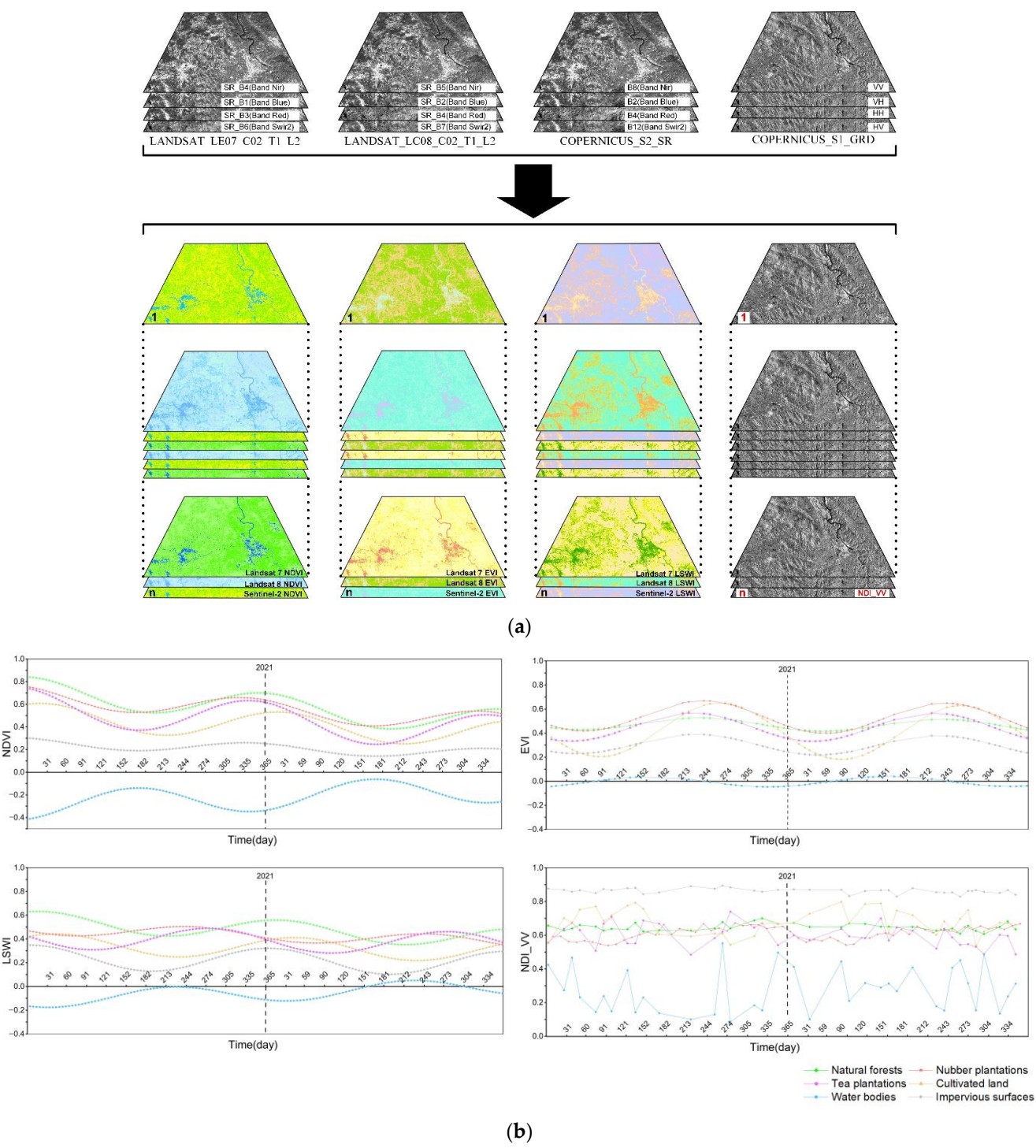

**Figure 7.** (**a**) The NDVI, EVI, LSWI and NDI_VV indexes of the four datasets were calculated and sorted according to time series to form a new dataset. (**b**) The HANTS algorithm was used to reconstruct each index curve.

To sum up, natural forests are the vegetation type that is most easily to be mixed with rubber, but the vegetation index of rubber in the defoliation and foliation period are significantly different from that of natural forests. Composite images of key phenological

window periods of rubber were used as input features for the Random Forest classification algorithm to avoid the influence of feature redundancy on the classification results. With the help of GEE platform, the phenological period was further subdivided by median synthesis (Table 4), while rubber defoliation during dry-hot periods in the dry season accounted for 73.87% of the annual defoliation [26]. Therefore, FVC [61] and Tasseled Cap Transformation (TCT) are added to the study to obtain the brightness (TCT-BRI), greenness (TCT-GRE) and wetness (TCT-WET) components [62–64] to highlight the leaf fall characteristics during this period. Table 5 displays the transformation coefficients [65] used in our study.

**Table 5.** Multispectral imagery Tasseled Cap Transformation (TCT) coefficients.

| Bands / Features | Blue | Green | Red | NIR | SWIR 1 | SWIR 2 |
|---|---|---|---|---|---|---|
| TCT-BRI | 0.0822 | 0.1360 | 0.2611 | 0.3895 | 0.3882 | 0.1366 |
| TCT-GRE | −0.1128 | −0.1680 | −0.3480 | 0.3165 | −0.4578 | −0.4064 |
| TCT-WET | 0.1363 | 0.2802 | 0.3072 | −0.0807 | −0.4064 | −0.5602 |

### 2.6. Random Forest (RF) Algorithm

Random Forest algorithm is a machine learning algorithm that can predict hundreds of explanatory variables, and it employs decision trees as units to aggregate numerous decision trees for classification, allowing for the categorization of vast quantities of higher-dimensional data [66]. Compared with other classification algorithms, Random Forest algorithm has the advantage of efficient training and less prone to overfitting. Furthermore, the algorithm implicitly includes discriminant weights for the classification effect of each metric to highlight features those are advantageous for classification [40].

### 2.7. Accuracy Assessment

On the basis of validation sample data (Figure 3), a confusion matrix was utilized to compute OA, PA, UA and Kappa [67]. The four evaluation indexes were utilized to examine the outcomes of rubber plantations identification. The OA and Kappa indices were used to assess the overall score accuracy, while PA and UA were used to assess the misclassification and omission errors among the LULC types [68], which were computed as follows:

$$PA = \frac{n_{ii}}{n_{\bullet i}} \times 100\% \tag{1}$$

$$UA = \frac{n_{ii}}{n_{i\bullet}} \times 100\% \tag{2}$$

$$OA = \frac{\sum\limits_{i=1}^{m} n_{ii}}{n} \times 100\% \tag{3}$$

$$Kappa = \frac{p_0 - p_e}{1 - p_e}$$
$$(p_0 = OA, \; p_e = \frac{\sum\limits_{i=1}^{q} n_{i\bullet} \times n_{\bullet i}}{n^2}) \tag{4}$$

where $i$ refers to the pixels; $n_{ii}$ is the total number of pixels; $m$ is the total number of diagonal pixels of the confusion matrix; $q$ is the number of classes in the confusion matrix; $n_{i\bullet}$ is the sum of row pixels of a class in the confusion matrix; $n_{\bullet i}$ is the sum of column pixels of a class in the confusion matrix.

### 2.8. Post-Classification Processing

The Random Forest classification algorithm is based on pixel-by-pixel classification; hence, the 'Salt and Pepper' phenomenon is difficult to avoid (when the pixels within a single land type are identified as other classes) [69]. In order to reduce the influence of noise on the classification results, we adopted the neighborhood mean filter algorithm,

and sets a 3 × 3 median filter kernel to smooth and denoise the final rubber plantations identification results to eliminate the impact.

## 3. Results

### 3.1. Phenological Characteristics of Rubber Plantations in XSBN

Rubber originally grew in evergreen broad-leaved forests in tropical areas; when it was transplanted to XSBN, China, in order to adapt to the low temperature in winter in this area, a unique phenomenon of leaf fall occurred. The spectral information of rubber plantations and other LULC classes during defoliation and foliation stages differed significantly (Figure 8).

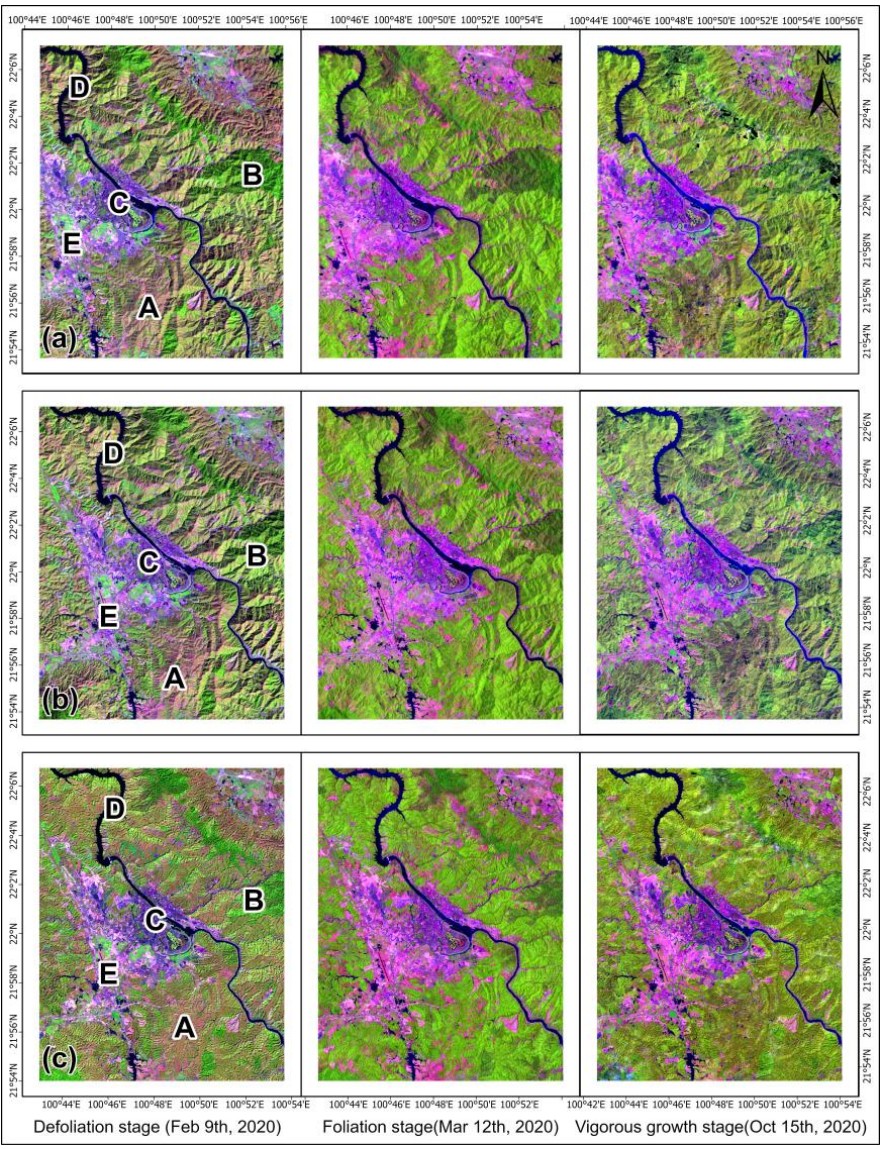

**Figure 8.** False color composite images of (**a**) Landsat-7/ETM+ (R/G/B = SR_B5/B4/B3), (**b**) Landsat-8/OLI (R/G/B = B6/B5/B4) and (**c**) Sentinel-2 MSI (R/G/B = B11/B8/B4) in the defoliation, foliation and vigorous growth stage of rubber plantations. As can be seen from the small area images, the spectral characteristics of the rubber plantations in the defoliation and foliation stages were significantly different from those of other land use/cover types, and the rubber plantations and the natural forests were easily mixed in the vigorous growth stage. (A) Natural forests, (B) rubber plantations, (C) impervious surfaces, (D) water bodies, (E) cultivated land are marked on the images.

Figure 8 shows, from left to right, the three-band false-color composite images of (a) Landsat-7/ETM+, (b) Landsat-8/OLI and (c) Sentinel-2 MSI, in the defoliation, foliation and growth stages of the rubber plantations. Rubber plantations in the defoliation stage seem shading yellow and may be distinguished from the deep green of natural forests, but both natural forests and rubber plantations in the foliation stage are green but vary in tone. The most difficult period to distinguish them is the period of vigorous growth; since both are shown in dark green, it is difficult to distinguish them only by spectral characteristics. Therefore, spectral reflectance differences in different phenology periods are critical in distinguishing rubber plantations from other easily mixed LULC types.

### 3.2. Accuracy Comparison of Different Classification Schemes

Accuracy of the five classification schemes are shown in Table 6. The table demonstrates that the classification accuracy derived from different feature combinations vary a lot. In CS 1, the composite images of NDVI and EVI were used as input features for classification. In CS 2, LSWI, elevation and slope were added on the basis of CS 1. In CS 3, brightness, greenness and wetness are added on the basis of CS 2. In CS 4 and CS 5, FVC and NDVI_VV index are gradually added on the basis of the previous classification scheme.

**Table 6.** Accuracy statistics of different classification schemes (%).

|  |  | Water Bodies | Impervious Surfaces | Tea Plantations | Cultivated Land | Natural Forests | Rubber Plantations | OA | Kappa |
|---|---|---|---|---|---|---|---|---|---|
| CS 1 | PA | 100 | 92.0 | 61.1 | 94.1 | 87.5 | 92.4 | 87.4 | 0.82 |
|  | UA | 100 | 94.8 | 76.3 | 88.2 | 87.5 | 88.0 |  |  |
| CS 2 | PA | 100 | 91.3 | 61.4 | 96.5 | 87.3 | 93.9 | 88.4 | 0.83 |
|  | UA | 100 | 93.3 | 79.2 | 94.2 | 87.3 | 88.0 |  |  |
| CS 3 | PA | 100 | 94.2 | 60.0 | 96.9 | 88.2 | 93.9 | 88.5 | 0.84 |
|  | UA | 98.4 | 92.9 | 79.1 | 95.3 | 86.3 | 88.2 |  |  |
| CS 4 | PA | 100 | 94.2 | 60.0 | 96.4 | 88.5 | 95.1 | 89.0 | 0.84 |
|  | UA | 98.4 | 92.9 | 83.7 | 95.0 | 87.1 | 88.0 |  |  |
| CS 5 | PA | 100 | 94.9 | 62.7 | 96.5 | 89.1 | 95.2 | 90.0 | 0.86 |
|  | UA | 98.4 | 92.9 | 84.4 | 95.3 | 88.2 | 88.8 |  |  |

Compared with the other four classification schemes, the OA and Kappa of CS 1 are the lowest, which shows that the rubber plantations cannot be effectively identified only by describing the two spectral index features of NDVI and EVI. After adding LSWI and Slope based on CS 2, the OA and Kappa coefficient of CS 3 increased by 1% and 0.01, respectively, compared with CS 1, and the PA of the rubber plantations increased by 1.5%, indicating that LSWI can reduce the mix-classification between rubber plantations and other land types. After we added FVC, TCT-BRI, TCT-GRE and TCT-WET features to CS 3, the OA and Kappa coefficient of CS 4 increased by 1.6% and 0.02, respectively, and the PA of the rubber plantations increased by 1.2% compared with CS 2; UA has not changed. CS 5 had the most input features included in the classification and the best final classification accuracy, with OA and Kappa coefficient values of 90% and 0.86, respectively.

By comparing CS 3, CS 4 and CS 5 (Figure 9), CS 3 has poor recognition results, with mixed classification between rubber plantations and impervious surfaces, cultivated land and natural forests. Meanwhile, it is easy to generate sporadic and fragmented patches, which is quite different from the real distribution of ground objects. The optimal combination of CS 5 (Type1 + Type2 + Type3 + Type4 + Type5) with the addition of NDI_VV index provides the most accurate classification results and efficiently reduces misclassification and omission between different categories.

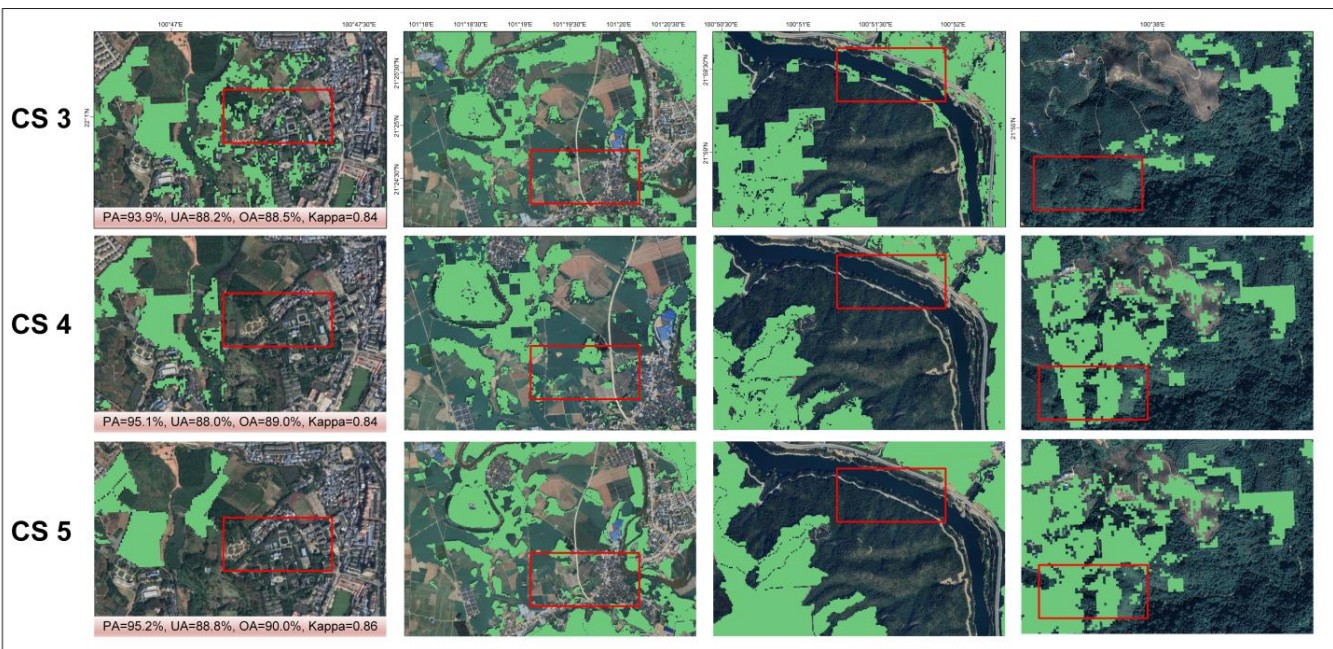

**Figure 9.** The classification results for CS 3, CS 4 and CS 5 rubber plantations are shown on a comparison map. The green portion is the rubber plantations identification result, whereas the bottom image is a high-resolution remote sensing image from Google Earth.

### 3.3. Accuracy Evaluation of Optimal Classification Scenarios

Table 7 shows the CS 5 accuracy evaluation confusion matrix, from which we can see that: (i) there is a mixed classification phenomenon between rubber plantations and tea plantations, cultivated land and natural forests, and the classification accuracy of tea plantations is relatively low, mainly because the spectral and textural features of tea plantations and rubber plantations are very similar. Although the phenology period obtained based on the time series curve can effectively distinguish most tea plantations, the broken terrain and mixed vegetation situation increases the difficulty and uncertainty of identification. (ii) With the exception of tea plantations, the PA and UA values for all LULC types are greater than 88%, with the PA values for water bodies, impervious surfaces, cultivated land and rubber plantations above 94%, which could achieve better classification results. (iii) For CS 5, the OA is 90.0%, Kappa coefficient is 0.86 and the PA and UA of rubber plantations exceed 85%, which satisfies the spatial analysis and practical application requirements.

**Table 7.** CS 5 accuracy evaluation confusion matrix (2020).

| Predicted Value / True Value | Water Bodies | Impervious Surfaces | Tea Plantations | Cultivated Land | Natural Forests | Rubber Plantations |
|---|---|---|---|---|---|---|
| Water bodies | 61 | 0 | 0 | 0 | 0 | 0 |
| Impervious surfaces | 0 | 131 | 1 | 0 | 0 | 0 |
| Tea plantations | 0 | 3 | 178 | 3 | 27 | 73 |
| Cultivated land | 1 | 5 | 0 | 245 | 0 | 3 |
| Natural forests | 0 | 0 | 6 | 0 | 285 | 29 |
| Rubber plantations | 0 | 2 | 26 | 3 | 11 | 835 |
| PA (%) | 100 | 94.9 | 62.7 | 96.5 | 89.1 | 95.2 |
| UA (%) | 98.4 | 92.9 | 84.4 | 95.3 | 88.2 | 88.8 |
| OA (%) | 90.0 | | | Kappa | | 0.86 |

In order to further verify the stability of the accuracy of the CS 5, the sample data were randomly divided into 10 sections, with 7 sections used to train the model and

3 sections used to test the accuracy after classification. The cross-validation was repeated 10 times to obtain the UA, PA (Figure 10), OA and Kappa (Figure 11) for rubber plantations identification in 2014, 2016, 2018 and 2020.

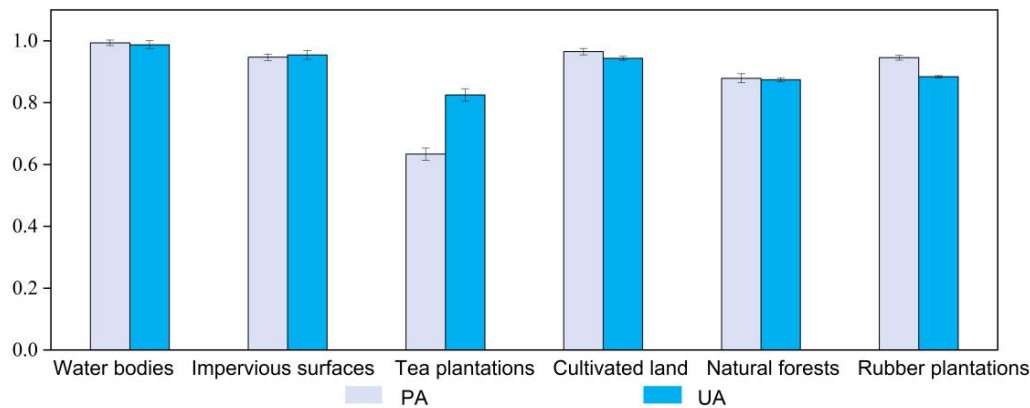

**Figure 10.** Results of cross-validation of PA and UA in rubber plantations in 2014, 2016, 2018 and 2020 (10 times of cross-validation for PA and UA values in each period).

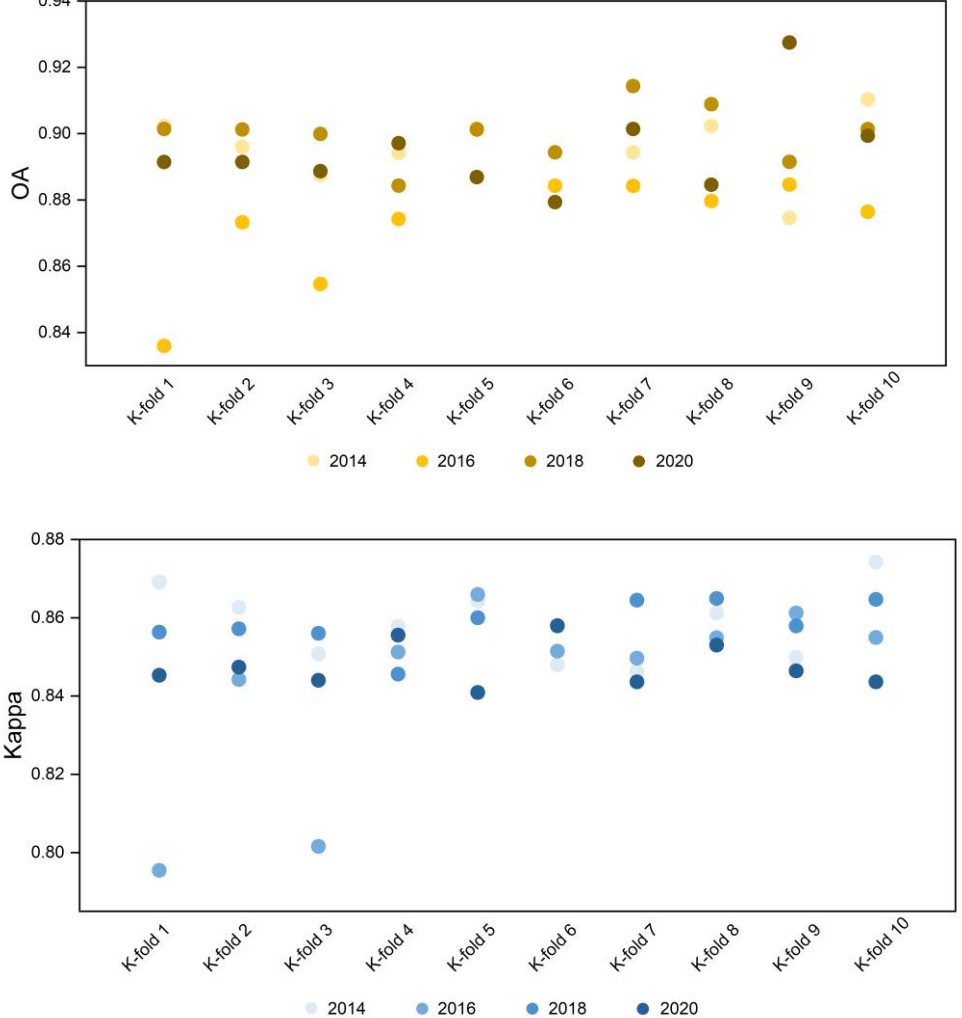

**Figure 11.** Results of cross-validation of OA and Kappa in rubber plantations in 2014, 2016, 2018 and 2020 (10 times of cross-validation for OA and Kappa values in each period).

Figure 10 shows that after 40 cross-validations, the PA and UA of water bodies, impervious surfaces, cultivated land, natural forests and rubber plantations can be stabilized at more than 85%, with the UA values varying the least and being the most stable. The values of PA and UA in tea plantations were lower (over 60% and 80%, respectively), and the fluctuation was shape. Figure 11 shows that the OA and Kappa coefficient values fluctuate between 0.82–0.94 and 0.82–0.88, respectively. The OA value is stable in the range of 0.88–0.90 and the Kappa coefficient in the range of 0.84–0.86, which indicates the stability and dependability of the CS 5.

Figure 12 shows the spatial distribution of XSBN rubber plantations in 2014, 2016, 2018 and 2020 obtained by CS 5. The total area of which are 4601.52 km$^2$, 4691.50 km$^2$, 4527.75 km$^2$ and 4199.28 km$^2$, respectively, showing a trend of increasing first and then decreasing. The identification results in 2018 are consistent with the Third National Land Resource Survey (sub-meter image combined with manual visual interpretation). As can be seen from the figure, the rubber plantations are mainly distributed in Jinghong and Mengla in the central and eastern regions, and relatively less in Menghai in the west. In this study, three typical rubber plantations planting areas were randomly selected according to different geomorphological characteristics and compared with Google Earth high-resolution remote sensing images. The comparison results show that the rubber plantations in these three areas can be accurately identified, and the identified boundary information is in good agreement with the image, confirming the accuracy and effectiveness of the method.

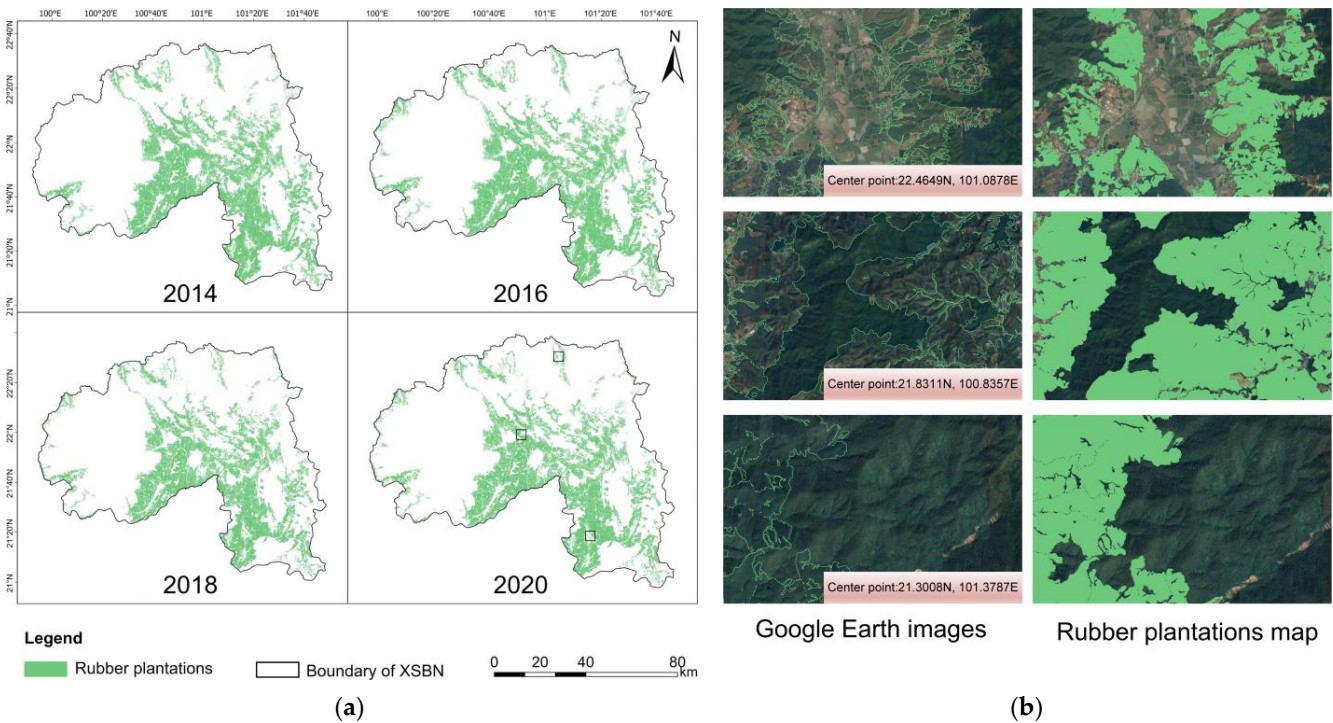

**Figure 12.** (**a**) Remote sensing identification results of four rubber plantations in 2014, 2016, 2018 and 2020 and (**b**) a typical planting area of rubber plantations in the north, south and central areas were selected for comparison with Google Earth high-resolution images.

## 4. Discussion

### 4.1. Characteristics of the "Stratified Sampling + Non-Homogenous Data Voting" Method of Sample Selection

How to effectively obtain accurate and reliable sample data has been one of the difficulties and challenges in land use and land cover classification [70], especially for plateau mountainous tropical rainforest areas with broken terrain and mixed vegetation (such as XSBN). In this study, we proposed a method of "stratified sampling + non-homogenous data voting" by using open, free and available 10 m spatial resolution LULC datasets. After

obtaining the categories existing in the non-homologous dataset, the selected range of the final sample points was determined by voting, which effectively reduced the error and uncertainty of determining sample points from a single data source. Compared with the traditional way of collecting samples in the field, this method is not only efficient and convenient, but also objective and not affected by human. However, the limitation of this method is that for tropical areas that do not have the characteristics of mountainous vertical zones, the recognition accuracy of rubber forests will be greatly limited because of the absence of the defoliation phenomenon.

From the identification results of rubber plantations and the research results of local experts, the method proposed in this study provides reliable results for the spatial distribution pattern of rubber plantations and is potentially transferable to other mountainous areas as a robust approach for rapid monitoring of rubber plantations.

### 4.2. Uncertainty

Using the GEE platform, based on the Random Forest classification algorithm, the spatial distribution information maps of rubber plantations in XSBN in 2014, 2016, 2018 and 2020 were drawn for the first time on a regional scale with a spatial resolution of 10 m. We have improved the spatial resolution and mapping accuracy to a certain extent, and can capture more fine-grained ground object information. However, there are some uncertainties due to data quality and availability, subtle differences in phenological periods for different vegetation cover types and the designed algorithm. First, affected by geographic location and climate conditions, the availability of L7_SR, L8_SR and S2_SR is the key to accurately identify rubber plantations. Although the GEE platform has embedded cloud removal algorithms for various datasets, each algorithm is different in different regions. Mistakes and omissions still occur and how to design appropriate cloud removal algorithms for different terrains and areas with dense cloud cover needs further discussion and research. Besides, although high classification accuracy can be obtained for rubber plantations identification, there are still planting patterns of intercropping rubber plantations and cash crops (rice, corn, pineapple or banana) in the study area, which will cause remote sensing identification results to overestimate or underestimate the real planting area of rubber plantations [34]. Third, there are obvious differences in spectral characteristics between XSBN rubber plantations and other land types in defoliation and foliation period by remote sensing images, which was consistent with the conclusions of previous local studies [29,34,35,71,72]. However, through on-the-spot investigation, we find that the phenology periods of rubber plantations in different years were slightly different, and the phenology periods of different locations in the same area were also inconsistent. The reason for this phenomenon may be related to topography, altitude and planting varieties of rubber plantations. Finally, the RF algorithm is based on pixel-by-pixel classification, and it is difficult to avoid the 'salt and pepper' phenomenon. Although the post-processing method was adopted to optimize the classification results, these effects are still the main problems faced by pixel classification. The introduction of object-oriented ideas and the combination of machine learning algorithms will be the focus of future research.

Previous studies have doubted and addressed the negative impact of commercial monoculture rubber expansion on carbon storage, deforestation and fragmentation impacts on biodiversity and ecosystem services. Rubber provides an important cash income to local smallholder farmers indeed, but in the perspective of capacity to support biodiversity, ecosystem services and human well-being, socio-ecological solutions are required to combat degradation and promote restoration at regional and landscape scales. A key challenge in making decisions regarding the rubber expansion management is its complexity and intractability. Tropical forests support a huge fraction of global terrestrial biodiversity and account for 25% of the terrestrial carbon pool. Decisions need to consider the stakeholders, collaborations and mutual interactions among different roles (e.g., governments, scientists) in such a diverse region as XSBN. Through the shared understanding of the process and of challenges in rubber expansion, such as forest ecology, carbon stocks, the use of technology

in management and the short and long run profitability, we would finally be able to achieve sustainable development goals.

## 5. Conclusions

This study integrates multi-spectral and synthetic aperture radar data. The Landsat-7/ETM+, Landsat-8/OLI and Sentinel-2 MSI image datasets were reconstructed. Based on the time series curves of NDVI, EVI and LSWI with 10 m spatial resolution, the phenological information of rubber plantations was determined by combining NDI_VV index. The characteristics of FVC, TCT-BRI, TCT-GRE, TCT-WET and topography were further obtained, and five classification schemes were constructed. The Random Forest classification algorithm was used for classification, and the spatial distribution and dynamic change maps of rubber plantations in XSBN in 2014, 2016, 2018 and 2020 were drawn. The classification results were verified by OA, Kappa coefficient, PA and UA. The main conclusions are as follows:

(1) By using available, free LULC datasets and Google Earth high-resolution images, the sample selection process was optimized using the method of "stratified sampling + non-homogeneous data voting", which effectively solved the problem of field samples in plateau mountainous areas. Research papers with an insufficient number of samples, often due to the high difficulty in obtaining them, are prone to errors and omissions.

(2) Five classification scenarios were developed for rubber plantations throughout the phenology period by integrating NDVI, EVI, LSWI, NDI_VV, TCT-BRI, TCT-GRE, TCT-WET and FVC composite images, slope and elevation data. Compared to the other scenarios, the addition of the NDI_VV index may significantly minimize the misclassification of rubber plantations and tea plantations while improving accuracy, which indicates the enormous potential of radar data in distinguishing tree species of varying heights.

(3) The four accuracy evaluation indexes of UA, PA, OA and Kappa coefficient derived from CS 5 were cross-validated, and the result indicated that the method proposed provides reliable results on spatial distribution of rubber in the fragmented terrain and mixed vegetation environment of highland mountainous regions, and is potentially transferable to other similar areas as a robust approach for rapid monitoring of rubber plantations.

**Author Contributions:** G.C. designed the entire framework and contributed significantly to the data collection; Z.L. and R.T. conducted experiments and wrote the manuscript; Q.W. has given many valuable suggestions for improving and modifying this paper. Y.W., J.Z. and J.F. contributed extensively to data processing. All the authors were involved in result analysis and discussion. All authors have read and agreed to the published version of the manuscript.

**Funding:** This research was supported by the Basic Research Project of Yunnan Province (Grant No. 202101AU070161) and the Strategic Priority Research Program of Chinese Academy of Sciences (Grant No. XDA26050301-01).

**Data Availability Statement:** All data used in this study are detailed in the manuscript.

**Acknowledgments:** We acknowledge the platform provided by Faculty of Land Resource Engineering of Kunming University of Science and Technology. We also want to thank the editor and reviewers for their insightful contributions and suggestions.

**Conflicts of Interest:** The authors declare no conflict of interest.

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
