# Peer review of "Identification of Rubber Plantations in Southwestern China Based on Multi-Source Remote Sensing Data and Phenology Windows"

_remotesensing, doi:10.3390/rs15051228_

Round 1

Reviewer 1 Report

see attached file

Author Response

Response to Reviewer 1 Comments

Dear editor and reviewer:

Thank you for your letter and for the reviewer’s first round comments concerning our manuscript (manuscript ID: remotesensing-2162680). Those comments are very valuable and helpful for revising and improving our paper, as well as the important guiding significance to our researches. We have studied comments carefully and have made correction which we hope meet with approval. Revised portion are marked in the paper. The main corrections in the paper and the responds to the reviewer’s comments are listed as follows.

Reviewer #1:

This paper provides an example of processing multi-sensor, multi-date imagery to identify land cover types in a tropical area and more precisely to monitor rubber plantations in this area from 2014 to 2020. Google Earth Engine is used to gather high spatial resolution optical data (Landsat 7/8, Sentinel2) and SAR data (Sentinel-1) and to obtain image time-series at a common 10m spatial resolution.

Several spectral indices, ancillary data (slope, elevation) and a normalized index of SAR data are used as inputs into RANDOM FOREST (RF) classifier. Five classification schemes are evaluated.

Point 1: They are mentioned at Figure 4, but it would be better to recall these 5 schemes before discussion of results (paragraph 4.2).

Response 1: According to your suggestion, we have re-written section 4.2 with a more detailed description of the 5 schemes before discussion the results.

Point 2: Several vegetation indices are considered: NDVI, EVI, TCTGreenness, FVC : as they are combined with other features, the effectiveness of using multiple redundant vegetation indices is not obvious.

Response 2: Thank you for your valuable and helpful suggestions on the revision and improvement of our paper, which has important guiding significance for our research. In our paper, it can be seen from the time series analysis in Figure 7 b that the index (NDVI, EVI, LSWI and NDI_VV) corresponding to rubber plantations in different time periods is significantly different from that of other land object types (natural forests, cultivated land, tea plantations, water bodies, and impervious surfaces). We did this by constructing five classification schemes. From CS 1 to CS 5, although the OA and Kappa coefficient of all the land classes were not improved significantly, the observation Table 6, "Rubber plantations", shows that the PA and UA of the rubber plantations were increased from 92.4 % and 88.0 % in CS 1 to 95.2% and 88.8% in CS 5.

Point 3: Temporal profiles of NDVI, EVI, LSWI and NDVI_VV are produced, then FVC and TCT Components are added. Please specify clearly which time range is used to derive inputs for RF.

Response 3: We are sorry for the insufficient description of the specific timing of RF input features. The specific time corresponds to the column "Time Windows" in Table 4, and we also give a detailed explanation in lines 347-356 of the article.

Point 4: Classification: all seasons or winter period only?

Response 4: In our article, the time period for classification is primarily concentrated from January to June, and the specific time corresponds to the column "Time Windows" in Table 4.

Point 5: Questions:

- Table 5: are these coefficients the same for Landsat7, Landsat 8, Sentinel 2?

- FVC: how are obtained the values of NDVImin and NDVImax? 

Response 5: Thank you for your valuable and helpful questions about our article. Landsat-7 (ETM+), Landsat-8 (OLI), and Sentinel-2 satellites have different sensors. However, TCT relies on sensors. The images obtained by different sensors will have different TCTs. In our article, considering the actual situation of the study area and the quality of the images, the TCT coefficient of the Sentinel-2 image was finally adopted.

In our paper, the pixel binary model is used to estimate FVC. NDVImin and NDVImax refer to the minimum NDVI value and the maximum NDVI value in the study area, respectively. We take NDVI values with histogram cumulative frequency of 5 % and 95 % as NDVImin and NDVImax.

Reviewer 2 Report

The present manuscript “Identification of Rubber Plantations in Southwestern China Based on Multi-source Remote Sensing Data and Phenology Windows” presents significant results for the society and the scientific community. However, before recommending the present study for publication, some points need correction. Therefore, I am considering this manuscript for minor revisions, highlighting the following points:

1 – In my reading of the text, I detected spelling and grammatical errors, so I suggest to the authors a review of the writing of the manuscript by a fluent in the language.

2 – According to the rules of Remote Sensing, only 300 words are allowed in the abstract, the authors present 383 words. The abstract must be synthesized.

3 – The keywords must be presented in alphabetical order.

4 – The keyword “Multi-source remote sensing data” is too long, I recommend authors remove it or replace it with another one.

5 – Where authors write “Landsat8”, rewrite to “Landsat-8/OLI or Landsat 8/OLI”, and make these changes throughout the body of the text.

6 – The same should be done for “Landsat7”, rewrite to “Landsat-7/ETM + or Landsat 7/ETM +”. Make these changes throughout the body of the text.

7 – In Figure 1, remove the latitude and longitude from the base and the right side of the maps.

8 – The scale of one of the maps in Figure 1 is wrong, the authors must correct it.

9 – Some parts of the text need a formatting review, such as lines 220 to 225, they are not justified with the margin of the document.

10 – Insert the latitude and longitude of the areas in Figure 9.

11 – In Figure 12, remove the latitude and longitude from the base and the right side of the maps.

12 – Based on the results presented, the authors should flesh out the text presented in the discussions.

Author Response

Response to Reviewer 2 Comments

Dear editor and reviewers:

Thank you for your letter and for the reviewers’ first round comments concerning our manuscript (manuscript ID: remotesensing-2162680). Those comments are very valuable and helpful for revising and improving our paper, as well as the important guiding significance to our researches. We have studied comments carefully and have made correction which we hope meet with approval. Revised portion are marked in the paper. The main corrections in the paper and the responds to the reviewer’s comments are listed as follows.

Reviewer #2:

The present manuscript “Identification of Rubber Plantations in Southwestern China Based on Multi-source Remote Sensing Data and Phenology Windows” presents significant results for the society and the scientific community. However, before recommending the present study for publication, some points need correction. Therefore, I am considering this manuscript for minor revisions, highlighting the following points:

Point 1: In my reading of the text, I detected spelling and grammatical errors, so I suggest to the authors a review of the writing of the manuscript by a fluent in the language.

Response 1: We are very sorry for our incorrect using of spelling and grammar, we have checked the whole manuscript and rectified them.

Point 2: According to the rules of Remote Sensing, only 300 words are allowed in the abstract, the authors present 383 words. The abstract must be synthesized.

Response 2: We have reformulated the Abstract section according to reviewer’s suggestion.

Point 3: The keywords must be presented in alphabetical order.

Response 3: We have rearranged the keywords in alphabetical order.

Point 4: The keyword “Multi-source remote sensing data” is too long, I recommend authors remove it or replace it with another one.

Response 4: According to your suggestion, we deleted “Multi-source remote sensing data” in the keyword, and replaced with “identification”.

Point 5: Where authors write “Landsat8”, rewrite to “Landsat-8/OLI or Landsat 8/OLI”, and make these changes throughout the body of the text.

Response 5: We are very sorry for our incorrect writing and we have corrected it

Point 6: The same should be done for “Landsat7”, rewrite to “Landsat-7/ETM + or Landsat 7/ETM +”. Make these changes throughout the body of the text.

Response 6: We are very sorry for our incorrect writing and we have rectified it.

Point 7: In Figure 1, remove the latitude and longitude from the base and the right side of the maps.

Response 7: Considering your suggestion, we made some revisions in Figure 1.

Point 8: The scale of one of the maps in Figure 1 is wrong, the authors must correct it.

Response 8: We are very sorry for our negligence of the scale in Figure 1 and we have made correction according to your comments.

Point 9: Some parts of the text need a formatting review, such as lines 220 to 225, they are not justified with the margin of the document.

Response 9: We are very sorry for the formatting issues, we have made corrections in the sentence involved.

Point 10: Insert the latitude and longitude of the areas in Figure 9.

Response 10: Considering your suggestion, we made some revisions in Figure 9.

Point 11: In Figure 12, remove the latitude and longitude from the base and the right side of the maps.

Response 11: We have made corresponding revisions in Figure 12.

Point 12: Based on the results presented, the authors should flesh out the text presented in the discussions.

Response 12: We have rewritten this part (Discussion section) according to the Reviewer’s suggestion.

Reviewer 3 Report

The Manuscript entitled “Identification of Rubber Plantations in Southwestern China Based on Multi-source Remote Sensing Data and Phenology Windows’’ did a detailed investigation with the main question of how to solve the difficulty and uncertainty faced in the identification of rubber plantations in Southwestern China. For that it proposed a classification method in combination with multi-source phenological characteristics .

This study adds the option of using multiple satellite data to solve the uncertainty of remote sensing identification with a classification method that combines multi-source phenological characteristics and a random forest algorithm. I consider the topic original and relevant in its field as it addresses the specific gap that a single satellite sensor cannot establish completed time series data. The authors integrated Landsat 7, 8, and Sentinel 2 optical images to construct time-series data for rubber plantations phenology on the GEE cloud platform, and then propose a pixel-based classification method integrated phenology window. The authors have concluded the work in brief but maintained the consistency of the aims. The main question posed in the article is fulfilled in the conclusion. In a nutshell, this is a good work, the language is okay but needs some revision before its consideration. My suggestions are mentioned below:

In the abstract, please re-write the 1st sentence, do not start it with ‘As’.

In introduction, please re-write last 3 paragraphs, if possible, merge them. Please do not start the final paragraph of introduction with ‘In conclusion’. There should be a clear aim/ hypothesis of this study. Further, there should be very distinct objective(s). During discussion, you have to clarify you have defended your objective(s).

L 120: please explain the term ‘Phenology Window’

In materials and methods, L 167: Complete data set title is unnecessary as it looks lengthy (For example LANDSAT_LE07_C02_T1_L2 (L7_SR), LANDSAT_LC08_C02_T1_L2 167 (L8_SR)).

Justify why you have used ‘stratified sampling + non-homogenous data voting’?

Please delete the URL from Table 2, it is too lengthy for a table.

Mention references for all equations.

Why are there three small boxes in the 2020 map within Figure 12 (a)? Please check.

Author Response

Response to Reviewer 3 Comments

List of Responses

Dear editor and reviewer:

Thank you for your letter and for the reviewer’s first round comments concerning our manuscript (manuscript ID: remotesensing-2162680). Those comments are very valuable and helpful for revising and improving our paper, as well as the important guiding significance to our researches. We have studied comments carefully and have made correction which we hope meet with approval. Revised portion are marked in the paper. The main corrections in the paper and the responds to the reviewer’s comments are listed as follows.

Reviewer #3:

The Manuscript entitled “Identification of Rubber Plantations in Southwestern China Based on Multi-source Remote Sensing Data and Phenology Windows’’ did a detailed investigation with the main question of how to solve the difficulty and uncertainty faced in the identification of rubber plantations in Southwestern China. For that it proposed a classification method in combination with multi-source phenological characteristics.

This study adds the option of using multiple satellite data to solve the uncertainty of remote sensing identification with a classification method that combines multi-source phenological characteristics and a random forest algorithm. I consider the topic original and relevant in its field as it addresses the specific gap that a single satellite sensor cannot establish completed time series data. The authors integrated Landsat 7, 8, and Sentinel 2 optical images to construct time-series data for rubber plantations phenology on the GEE cloud platform, and then propose a pixel-based classification method integrated phenology window.      

The authors have concluded the work in brief but maintained the consistency of the aims. The main question posed in the article is fulfilled in the conclusion. In a nutshell, this is a good work, the language is okay but needs some revision before its consideration. My suggestions are mentioned below:

Point 1: In the abstract, please re-write the 1st sentence, do not start it with ‘As’.

Response 1: We are very sorry for our incorrect using of spelling and grammar, we have checked the whole manuscript and rectified them, and we have reformulated the Abstract section according to reviewer’s suggestion, including the 1st sentence.

Point 2: In introduction, please re-write last 3 paragraphs, if possible, merge them. Please do not start the final paragraph of introduction with ‘In conclusion’. There should be a clear aim/ hypothesis of this study. Further, there should be very distinct objective(s). During discussion, you have to clarify you have defended your objective(s).

Response 2: Based on your suggestions, we made some revisions in the Introduction and Discussion part, which can be found in the revised marks.

Point 3: L 120: please explain the term ‘Phenology Window’.

Response 3: We are sorry about the insufficient description of the concept of ‘Phenology Window’. Since vegetation growth has distinct and relatively consistent time variations. Vegetation also has different physiological and appearance characteristics during different seasons and growth periods, and the time corresponding to this feature is referred as the "Phenology Window." Rubber was originally an evergreen broadleaf forest growing in the tropics. When it was transplanted to Xishuangbanna, China, in order to adapt to the low temperatures in winter, a unique defoliation phenomenon appeared, which became the phenological window period of rubber.

Point 4: In materials and methods, L 167: Complete data set title is unnecessary as it looks lengthy (For example LANDSAT_LE07_C02_T1_L2 (L7_SR), LANDSAT_LC08_C02_T1_L2 167 (L8_SR)).

Response 4: We have revised the data set title with commonly used title.

Point 5: Justify why you have used ‘stratified sampling + non-homogenous data voting’?

Response 5: The reason why we use ‘stratified sampling + non-homogenous data voting’ is because it has the following two advantages: (1) The selection range of the final sample points is determined by voting after the hierarchical extraction of the categories in the non-homologous data set, which effectively reduces the error and uncertainty of determining the sample points from a single data source. (2) When compared to the traditional method of collecting samples in the field, this method is not only faster and more convenient, but the selection process is objective and not influenced by human factors.

Point 6: Please delete the URL from Table 2, it is too lengthy for a table.

Response 6: Considering your suggestion, we made some revisions in Table 2.

Point 7: Mention references for all equations.

Response 7: According to your suggestion, we added corresponding references for all equations.

Point 8: Why are there three small boxes in the 2020 map within Figure 12 (a)? Please check.

Response 8: Those three small boxes in the 2020 is actually the locations of three typical zones of rubber trees in the following Figure 12 (b), presented just as examples.

Reviewer 4 Report

1.  Topic is very good. However, the contents related to the problem statement / rationale for the study is underdeveloped. Authors are not clear about the concept of Carbon fixation by Rubber tree. Rubber tree has more potential of carbon sequestration during the entire life. Therefore, it needs to be revised at all places, more specifically in the Abstract and Introduction, with a careful comparison and analysis while developing argument regarding it. This is the major issue in the problem as the conceptual framework has error.

2. Abstract is very long and needs to be shortened.

3. Study area section needs to be made sub-section of Methodology. For the Purpose, a main Section on Methodology be created and rest all should be made sub-sections including the study area, methods etc.

4. Provide methodological limitations.

5. It is good to see flow diagram. However, Figure 4 is very much confused. Revise Figure 4 to provide a clear flow diagram on methodological steps followed.

6. Discussion part is underdeveloped. There is a need to enrich it and particularly by adding an analysis in the context of problem/rationale / objective of the study, but after updating vide comment 1 above.

7. Revisit conclusion in the light of above comments # 1 & 6.   

Author Response

Response to Reviewer 4 Comments

Dear editor and reviewer:

Thank you for your letter and for the reviewer’s first round comments concerning our manuscript (manuscript ID: remotesensing-2162680). Those comments are very valuable and helpful for revising and improving our paper, as well as the important guiding significance to our researches. We have studied comments carefully and have made correction which we hope meet with approval. Revised portion are marked in the paper. The main corrections in the paper and the responds to the reviewer’s comments are listed as follows.

Point 1: Topic is very good. However, the contents related to the problem statement / rationale for the study is underdeveloped. Authors are not clear about the concept of Carbon fixation by Rubber tree. Rubber tree has more potential of carbon sequestration during the entire life. Therefore, it needs to be revised at all places, more specifically in the Abstract and Introduction, with a careful comparison and analysis while developing argument regarding it.

Response 1: Thank you for your valuable and helpful suggestions for revising and improving our paper, which provides important guiding significance to our researches. As the reviewer mentioned, cultivation of rubber trees could act as a carbon sink by sequestering carbon in biomass and indirectly in soils. But for Xishuangbanna, one of the most diverse ecosystems in China, is located within the Indo-Burma biodiversity hotspot. It comprises only 0.2% of China’s land area, it harbors nearly 16% of plant species, 21.7% of mammals, 36.2% of birds, 15% of amphibian and reptiles found in China. XB has been promoted as one of 32-biodiversity conservation priorities by the Ministry of Environment Protection emphasized in the 2011–2030 China Biodiversity Conservation Strategy and Action Plan. Geographically, it connects two global biodiversity hotpots, the Mountains of Southwest China and Indo-Burma. During the last two decades, XSBN has been largely transformed from biodiverse natural forests and mixed-use farms into commercial monoculture rubber plantations and many scholars have demonstrated the negative effect of this conversion. So in the manuscript before, we did emphasized the hazards over benefits. Based on your suggestion, we revised the whole paper, especially the sections with misunderstanding in the potential of rubber in carbon sequestration.

Point 2: Abstract is very long and needs to be shortened.

Response 2: Considering your suggestion, we have reformulated the Abstract section.

Point 3: Study area section needs to be made sub-section of Methodology. For the Purpose, a main Section on Methodology be created and rest all should be made sub-sections including the study area, methods etc.

Response 3: Considering your suggestion, we made some revisions in those sections, which can be found in the revised version with marks.

Point 4: Provide methodological limitations.

Response 4: We are sorry about the insufficient description of limitations of the method employed, the limitation of this method is that for tropical areas that do not have the characteristics of mountainous vertical zones, the recognition accuracy of rubber forests will be greatly limited because the absence of the defoliation phenomenon. so we added a sentence in the Discussion section, which can be found in the revised manuscript.

Point 5: It is good to see flow diagram. However, Figure 4 is very much confused. Revise Figure 4 to provide a clear flow diagram on methodological steps followed.

Response 5: Thank you for your valuable and helpful suggestions on the improvement of our paper. We are very sorry that the arrows in Figure 4 in our article may have caused you confusion. Each dotted wireframe in Figure 4 represents a step in the experiment. We designed five steps in the experiment to reproduce the entire experimental process in more detail. Therefore, we did not revise the flow chart in our article.

Point 6: Discussion part is underdeveloped. There is a need to enrich it and particularly by adding an analysis in the context of problem/rationale/objective of the study, but after updating vide comment 1 above.

Response 6: Considering your suggestion, we added an analysis in how our experiments fulfilling the objective of the study in the Discussion section.

Point 7: Revisit conclusion in the light of above comments # 1 & 6. 

Response 7: Thank you for your comments again and we have rewritten the conclusion based on the above suggestions.

Round 2

Reviewer 3 Report

I feel the correction/ improvement done by the authors are okay. I have no further problem with this manuscript.

Author Response

List of Responses

Dear editor and reviewer:

Thank you for your letter and for the reviewer’s second round comments concerning our manuscript (manuscript ID: remotesensing-2162680). Those comments are very valuable and helpful for revising and improving our paper. We have made correction based on your comments which we hope meet with approval. Revised portion are marked in the paper. The main corrections in the paper are listed as follows.

Reviewer #3: Comments and Suggestions for Authors

Point 1: I feel the correction/ improvement done by the authors are okay. I have no further problem with this manuscript. English language and style are fine/minor spell check required

Response 1: We are very sorry for our incorrect using of spelling and grammar, we have checked the whole manuscript and rectified them.

Reviewer 4 Report

I am comfortable with most of the replies except for response to comment # 1 regarding the conceptual framework and problem statement. Since the authors have mentioned monoculture aspect of Rubber trees, it needs to be emphasized in introduction for problem statement / contextual framework considering its negative impact on the native ecosystem. The same should also be reflected in discussion as well as conclusion parts. 

Author Response

List of Responses

Dear editor and reviewer:

Thank you for your letter and for the reviewer’s second round comments concerning our manuscript (manuscript ID: remotesensing-2162680). Those comments are very valuable and helpful for revising and improving our paper. We have made correction based on your comments which we hope meet with approval. Revised portion are marked in the paper. The main corrections in the paper are listed as follows.

Reviewer #3: Comments and Suggestions for Authors

Point 1: I feel the correction/ improvement done by the authors are okay. I have no further problem with this manuscript. English language and style are fine/minor spell check required

Response 1: We are very sorry for our incorrect using of spelling and grammar, we have checked the whole manuscript and rectified them.

Reviewer #4: Comments and Suggestions for Authors

Point 1: I am comfortable with most of the replies except for response to comment # 1 regarding the conceptual framework and problem statement. Since the authors have mentioned monoculture aspect of Rubber trees, it needs to be emphasized in introduction for problem statement / contextual framework considering its negative impact on the native ecosystem. The same should also be reflected in discussion as well as conclusion parts.

Response 1: Thank you for your insightful and helpful suggestions for revising our paper, we reformulated the Introduction and Discussion Section about monoculture structure of Rubber trees based on our understanding on your suggestions, which can be found in the revised version notes.

And we also took some typical Rubber plantations pictures in XSBN, which can describe the monoculture structure and leaf litter.
